# Muscle Strength and Balance as Mediators in the Association between Physical Activity and Health-Related Quality of Life in Community-Dwelling Older Adults

**DOI:** 10.3390/jcm11164857

**Published:** 2022-08-18

**Authors:** Marcelo de Maio Nascimento, Bruna R. Gouveia, Élvio Rúbio Gouveia, Pedro Campos, Adilson Marques, Andreas Ihle

**Affiliations:** 1Department of Physical Education, Federal University of Vale do São Francisco, Petrolina 56304-917, Brazil; 2LARSYS, Interactive Technologies Institute, 9020-105 Funchal, Portugal; 3Center for the Interdisciplinary Study of Gerontology and Vulnerability, University of Geneva, 1205 Geneva, Switzerland; 4Regional Directorate of Health, Secretary of Health of the Autonomous Region of Madeira, 9004-515 Funchal, Portugal; 5Saint Joseph of Cluny Higher School of Nursing, 9050-535 Funchal, Portugal; 6Department of Physical Education and Sport, University of Madeira, 9020-105 Funchal, Portugal; 7Department of Informatics Engineering and Interactive Media Design, University of Madeira, 9020-105 Funchal, Portugal; 8CIPER, Faculty of Human Kinetics, University of Lisbon, 1495-751 Lisbon, Portugal; 9ISAMB, Faculty of Medicine, University of Lisbon, 1649-020 Lisbon, Portugal; 10Department of Psychology, University of Geneva, 1205 Geneva, Switzerland; 11Swiss National Centre of Competence in Research LIVES—Overcoming Vulnerability: Life Course Perspectives, 1015 Lausanne, Switzerland

**Keywords:** ageing, quality of life, physical activity, lower limb strength, body balance, older adults

## Abstract

Lower extremity muscle strength (LEMS) and body balance (BB) are essential for older adults to maintain an upright posture and autonomously perform their basic activities of daily living. This study aimed to examine whether LEMS and BB mediate the relationship between physical activity (PA) and health-related quality of life (HRQoL) in a large sample of community-dwelling older adults. This is a cross-sectional study carried out with 802 individuals, 401 males and 401 females (69.8 ± 5.6 years), residents of the Autonomous Region of Madeira, Portugal. PA and HRQoL were assessed by the Baecke Questionnaire and e SF-36, respectively. LEMS was assessed by the Senior Fitness Test and BB by the Fullerton Advance Balance (FAB). The serial mediation pathway model pointed out that LEMS and BB partially mediated the association between PA and HRQoL in approximately 39.6% and 47%, respectively. The total variance in HRQoL explained by the entire model was 98%. Our findings may indicate the role that LEMS and BB play in the relationship between PA and HRQoL in the older population.

## 1. Introduction

Vulnerable aging is associated with a progressive decline in physiological [1], cognitive [2], and motor functions [3,4,5], making it difficult for older people to adapt to the environment in which they live. The motor functions that change due to vulnerable aging include lower extremity muscle strength (LEMS) and body balance (BB) deficits. Both are strongly associated in order to assure adequate postural control, i.e., it is necessary to have a moderate to high strength performance in the lower extremity parts of the body [6,7]. In advanced age, low LEMS and BB performance levels can lead the individual to frailty. Frailty is a multidimensional term used to indicate a geriatric syndrome characterized by reduced homeostatic reserves [8,9], responsible for exposing the older adult to an increased risk of negative health-related events (including acute chronic illness, disability, risk of falls, vulnerability or lack of strength and resilience, mental health, and mortality). In this context, moderate to high levels of physical activity (PA) are essential for older adults to present an adequate performance of LEMS and BB [10,11], consequently benefiting the performance of their basic activities of daily living (ADL) [12,13].

The function of the BB is to permanently stabilize the body, which is constantly disturbed by external and internal forces [14,15]. Thus, the BB maintains the center of gravity at the support base within the stability limits in static positions and dynamic situations, with minimal oscillation [16,17]. BB regulation is multifactorial, consisting of automatic reactions that cause an increase in muscle tone, avoiding a total imbalance [18]. The actions are monitored by the central nervous system (CNS), responsible for receiving postural information, analyzing/integrating, and quickly deciding which muscle output is most efficient to restore stability [19]. All postural synergy depends on the integration of sensory inputs performed by the afferent neural systems (e.g., visual, vestibular, and somatosensory) [20,21], in strict accordance with the efferent system (e.g., musculoskeletal) [22]. In this context, the synergy between the legs and trunk muscles is essential for older adults to present adequate levels of stable upright posture [19].

Consequently, due to age-related losses of skeletal muscle mass [23], there is a reduction in muscle strength [24]. Among older adults, sarcopenia, defined by loss of skeletal muscle mass and function, arises from multiple factors (e.g., cellular, neural, metabolic, hormonal) [25]. It is worth noting that the physical component of frailty is sometimes called sarcopenia [26], as it implies the loss of muscle mass and skeletal muscle strength. LEMS is a prerequisite for maintaining an upright posture and the effectiveness of immediate postural reactions in response to external disturbances [26,27]. Consequently, weakness or low muscle power limits a postural reaction, increasing the chance of losing the BB [28] and falling [13]. Moreover, it is worth mentioning that falls are a public health problem in old age, which can lead to fractures, days of hospitalization, and are responsible for one in three deaths of older adults [29].

The literature highlights that in the older population, there is a relationship between the low performance of LEMS and BB with reduced levels of PA [30]. In this sense, a low level of PA can generate limitations, affecting, among older adults, the performance of LEMS and BB. PA is the set of activities performed by the individual related to housework, sport, and leisure, estimating their energy expenditure [31,32]. Therefore, the higher the level of PA, the more active the older adult, and consequently, the greater the probability of having a high level of functional fitness, including adequate LEMS and BB performance [33]. On the other hand, low levels of PA are indicative of a sedentary lifestyle (low caloric expenditure) with a risk for changes in multiple physiological systems (e.g., neuromotor, musculoskeletal, cardiorespiratory) [34]. Consequently, this set of changes can negatively affect the perception of health-related quality of life (HRQoL) [35,36,37].

HRQoL is a multidimensional subjective construct, which refers to an individual’s situational perception about the set of satisfactions related to their daily lives, including physical, psychological, and social aspects [38]. When older adults realize that an increase in their PA levels has benefits for physical health (improvement in functional capacity), providing greater disposition and vitality [39], benefits for brain functions [40], as well as mental health (e.g., anxiety, depression, stress, mood) [41], gradually they feel more motivated to carry out their daily activities. This perception of well-being also motivates older adults to expand social relationships [42]. All this generates a positive health cycle in which PA improves HRQoL [43], and the improvement in HRQoL perception encourages the older adult to remain active and healthy [36].

The specialized literature recognizes the value of studying aging markers such as LEMS and BB [6,7,12,13,44] as a potential aid in the creation of strategies that can mitigate health problems and promote active [45] and healthy aging [46]. LEMS [47] and low BB performance [48] are an indicator of vulnerability [49], increasing the risk of falling [50] and mortality [51]. Moreover, all these factors are part of the term frailty and can negatively potentiate HRQoL [52,53]. Systematic review studies with meta-analysis [54,55] as well as experimental studies [6,27] highlighted the role of PA or physical exercise in the change in muscle coordination (balance) and muscle structure throughout life. However, to our knowledge, no study has associated LEMS and BB in the relationship between PA and HRQoL. Thus, considering that the levels of PA and HRQoL tend to decrease gradually with aging and that there is a direct relationship between both LEMS and BB, the present study aimed to examine in-depth whether LEMS and BB mediate the relationship between PA and HRQoL in a large sample of community-dwelling older adults.

## 2. Materials and Methods

### 2.1. Study Design and Participants

A cross-sectional analytical observational study was carried out with 802 individuals, 401 males and 401 females (69.8 ± 5.6 years). Participants were residents of different districts of the Autonomous Region of Madeira, Funchal, Portugal, recruited through direct contacts carried out by the principal investigator in day-care centers, nursing homes, cultural and sports clubs and associations, and residential and public places (e.g., open markets, municipal gardens, and churches). The study was also advertised in the daily newspaper, radio, and television. The inclusion criteria were: (1) Being community-dwelling, (2) age between 60 and 79 years, and (3) being able to walk independently. The exclusion criteria were: (1) Medical contraindications for submaximal physical exercise, according to the guidelines of the American College of Sports Medicine [56], and (2) not being able to understand or follow the investigation evaluation protocol. The Scientific Commission scientifically and ethically approved the study of the Department of Physical Education and Sports of the University of Madeira, the Regional Secretary of Social Affairs Committees, and received support from FCT (SFRH/BD29300/2006). All participants were informed about the procedures, and consent was provided to all participants before the assessments. This study included adherence to the declaration of Helsinki. Assessments were conducted in the Human Physical Growth and Motor Development Laboratory at the UMa by trained research personnel in 2009.

### 2.2. Data Collection

#### 2.2.1. Demographics and Health Profile

Through face-to-face interviews, participants reported the following sociodemographic and health information: gender, age, comorbidities, falls in the last 12 months, and the number and types of medication consumed daily. Trained field staff members conducted the interviews.

#### 2.2.2. Anthropometry

Body mass and height were measured using an anthropometric scale and a Welmy^®^ (London, UK) stadiometer coupled with 0.1 cm and 0.1 kg [57]. BMI was defined as (weight [kilograms])/(height [m]^2^).

#### 2.2.3. Physical Activity

PA was measured using the Baecke Questionnaire [58]; for validation, see, e.g., Gouveia et al. [59]. The questions investigate PA with a reference period of the last 12 months. The instrument is composed of three specific domains, namely: (1) Work/ housework (PA-work); (2) sports activities (PA-sport), only regular activities lasting at least one hour per week, and (3) free time activities (PA-leisure). In this study, the total physical activity level (PA-total) score was obtained by the following equation: PA-total = PA-work + PA-sport + PA-leisure/3.

#### 2.2.4. Health-Related Quality of Life

HRQoL was assessed using the Portuguese version [60] of the 36-item Short-Form Health.

Survey (SF-36 [61]). The SF-36 includes eight dimensions: physical functioning (PF), physical role (PR), bodily pain (BP), general health (GH), vitality (VT), social functioning (SF), emotional role (ER), and mental health (MH). By adding the scores, it is possible to analyze the HRQoL by two components: (1) Physical (PF + RP + BP + GH) and (2) Mental (VT + SF + ER + MH). Their scores range from 0 to 100 each. The present study considered the total SF-36 score obtained by adding the physical and mental components.

#### 2.2.5. Lower Extremity Muscle Strength

LEMS was assessed using the Senior Fitness Test (SFT) [62]. Following the study goal, we used the 30 s chair stand test to assess lower extremity muscle strength. The procedure was as follows: at the signal, the participant with arms crossed over the chest got up from a chair and returned to the original position (fully seated). The task continued for 30 s until the individual completed as many runs as possible. The score corresponded to the total number of executions performed correctly in 30 s. A detailed description of the test administration protocol, equipment used, and safety tips can be found [62].

#### 2.2.6. Body Balance

BB assessment was performed using the Fullerton Advanced Balance (FAB) scale [63]. The FAB is an instrument capable of measuring the older population’s multiple dimensions of static and dynamic balance. The protocol used includes 10 tasks: (1) Standing with feet together and eyes closed, (2) reaching out to pick up an object (pencil) held at shoulder height with the arm extended, (3) rotating 360° right and left, (4) step up and down a 15 cm bench, (5) walk-in tandem, (6) stand on one leg, (7) stand on foam with eyes closed, (8) long-jumping, (9) walking with head turned, and (10) recovering from an unexpected loss of balance. Each test item was scored on a 4-point ordinal scale (0–4), resulting in a maximum of 40 points. The FAB scale’s predictive validity concerning the risk of falling was previously presented [64]. A detailed description of the test administration protocol, equipment, and instructional video can be accessed [65].

#### 2.2.7. Covariates

In the present study, the following variables were assumed as possible confounding factors and, therefore, controlled in the serial analysis of mediation: sex, age, number of falls in the last 12 months, number and types of medications consumed daily, musculoarticular problems and BMI.

### 2.3. Statistical Analysis

The Kolmogorov–Smirnov test was initially applied to examine the data distribution. Thus, categorical variables were presented as frequencies and percentages, while continuous variables were presented as mean and standard deviation (SD). The main characteristics of the participants were compared using the chi-square test (categorical variables) and the unpaired Student’s *t*-test for independent samples (continuous variables). Regarding our main study goal, we conducted mediation analyses to examine whether LEMS and BB mediate the relationship between PA and HRQoL (see Figure 1 for a general illustration). It was considered a mediation or indirect effect when the causal effect of an independent variable X (physical activity) was able to predict the dependent variable Y (quality of life) mediated by *M*_1_ and *M*_2_ (LEMS and BB) [66]. A complete mediation would be observed if the simultaneous inclusion of the two objective measures *M*_1_ and *M*_2_ reduced the association between X and Y to zero. Therefore, a partial mediation would occur if, after the inclusion of *M*_1_ and *M*_2_, the association between X and Y became weaker. Furthermore, the indirect effect was considered significant when the confidence interval did not include zero. Our mediation hypotheses were tested using the bias-corrected bootstrap method with 5000 samples to calculate confidence intervals (95%). All analyses were processed by PROCESS v4.0 [67], a computational complement of the SPSS program. The significance level adopted for all analyses was α = 0.05.

## 3. Results

### 3.1. Main Characteristics of Participants

Table 1 presents the characteristics of the 802 participants. The division by sex was balanced, with half of the sample composed of women (69.75 ± 5.64 years old) and the other of men (69.87 ± 5.55 years old). The group had a low percentage of falls, 0.83 ± 1.60, and average daily consumption of types of medication of 3.57 ± 2.56. Regarding anthropometry, the mean height was 159.05 ± 8.69 cm, body weight was 74.77 ± 13.06 kg, and the mean BMI was 29.51 ± 4.34 kg/m^2^. Among the most common comorbidities were: hypertension (50.9%), visual impairment (61%), hearing problems (24.7%), and musculoarticular problems (5.7%). Regarding performance in the study variables, the average results were as follows: PA (7.30 ± 1.23), HRQoL (68.57 ± 17.96), LEMS (13.63 ± 4.14), and BB (30.53 ± 7.40).

### 3.2. Mediation Analysis

Figure 1 presents the results of the serial mediation analysis. The model obtained was significant, predicting the two variables F(5.000) = 38.7391, *p* < 0.001, R^2^ = 0.24. Model 1 was controlled for confounders (i.e., sex, age, falls, medication, musculoarticular problems, and BMI) and showed that PA (independent variable) had a positive and significant association with the LEMS mediator (*β* = 1.04, *t* (5.000) = 8.365, *p* < 0.001), and also with the BB mediator (*β* = 0.82, *t* (6.000) = 4.056, *p* = 0.001). The estimated association between both mediators (*m*_1_–*m*_2_) was positive and significant (*β* = 0.62, *t* (6.000) = 10.000, *p* < 0.001). Model 2 indicated significant and positive and significant associations between the LEMS mediator (*β* = 0.85, *t* (7.000) = 4.742, *p* < 0.001) and the BB mediator (*β* = 0.54, *t* (7.000) = 4.983, *p* < 0.001) with HRQoL (dependent variable). When the mediating variables (*m*_1_ and *m*_2_) were included, the path estimated by the model remained significant. Thus, the direct effect estimated by the model (*x*–*y*) indicated a positive and significant relationship between PA and HRQoL (*β* = 1.89, *t* (5.000) = 3.469, *p* = 0.006), and the total effect of the model (*x*–*y*) showed also a positive and significant relationship between PA and HRQoL (*β* = 3.56, *t* (5.000) = 3.469, *p* < 0.001). In our serial mediation path model, three effects were tested: (1) the indirect path through LEMS was significant (*β* = 0.05, 95% CI BCa = 0.351–0.861), (2) the specific indirect effect through BB was also significant (*β* = 0.292, 95% CI BCa = 0.0124–0.0509) and (3) the general indirect pathway from PA to HRQoL was partially mediated by LEMS and BB (*β* = 0.0231, 95% CI BCa = 0.0121–0.0358). These findings indicated that LEMS and BB were independent mediators of the positive effect that PA has on the perception of HRQoL in community-dwelling older adults. Finally, the proportion of the total effect of PA on HRQoL mediated by LEMS and BB was 98%. LEMS explained approximately 39.6% and BB 47% of the variance of the association between PA and HRQoL.

Figure 1 Mediation analysis: LEMS and BB in the relationship between PA and HRQoL.

## 4. Discussion

The present study examined whether LEMS and BB mediate the relationship between PA and HRQoL in a large sample of community-dwelling older adults. Our main finding was that when entering the performance results of the objective assessment LEMS and BB simultaneously as mediators, controlling for confounders (i.e., sex, age, falls, medication, musculoarticular problems, and BMI), the effects of the direct and total trajectory between PA and HRQOL (*x*–*y*) remained significant. Thus, we found that LEMS and BB partially mediated the association between PA and HRQoL in approximately 39.6% and 47%, respectively. These results suggested the role of lower muscle strength for postural control and the importance of adequate levels of LEMS and BB for older adults to achieve better levels of HRQoL. Finally, the total variance in HRQoL explained by the estimated model (as a whole) was 98%. To our knowledge, this study is the first to demonstrate this mediating pathway.

Our findings are consistent with the idea that, at advanced ages, higher PA levels can contribute to better physical function [5]. On the other hand, among older adults, a sedentary behavior (low level of PA), especially when prolonged, affects muscle physiology, decreasing muscle strength and functionality [68]. Consequently, the performance of all fitness capacities (e.g., cardiovascular endurance, flexibility, gait, coordination, body composition), including the maintenance or recovery of static and dynamic balance, is affected [7,19].

Due to the physiological changes of vulnerable aging, approximately 20% to 30% of skeletal muscle mass is lost between young adulthood and 80 years [69]. This can gradually affect BB. As a result, the size and number of muscle fibers, especially type II, decreases, rapidly declining muscle power [70]. Consequently, muscle strength loses an important prerequisite for performing or correcting accurate and rapid postural reactions in situations caused by external disturbances [11], causing a delay in postural correction. A possible cause of muscle strength loss in advanced age may be type II skeletal muscle fiber atrophy due to physiological aging [71]. One explanation for the impairment of the vasodilator function of muscle fibers is the progressive and gradual loss of capillaries [72]. On the other hand, a strategy to increase capillary supply is the regular practice of physical exercises, capable of increasing muscle blood flow [73]. In old age, a possible cause of muscle strength loss may be type II skeletal muscle fiber atrophy, which may be associated with decreased physiological reserve, known as frailty [26,52].

Damage to motor neuron functioning reduces maximum motor unit firing rates, impairing muscle power [74]. In turn, age-related physiological changes in muscle fibers affect the sensorimotor functions of different muscle groups (e.g., knee, plantar flexors, dorsal ankle) responsible for maintaining BB [22,75]. In an attempt to maintain an upright posture in imbalance situations, older adults usually apply three strategies sequentially [76]: ankle, hip, and step. The responsibility for controlling the action is the CNS, which emits stimuli (compensatory torques) to the hip towards the ankle, even increasing the speed of leg movements [77]. When the neuromuscular activation and the contractile properties of the fibers have suffered physiological decreases due to age, the older adult will need a high capacity to generate substantial compensatory mechanisms that allow the restoration of LEMS and BB. A measure capable of improving the strength of the LEMS and the performance of the BB is the participation in exercise programs that work the overload of the muscles and stimulate the increase of the speed of muscular contraction [78]. Therefore, it is also known that conventional strength training, when associated with power training (fast concentric phase), can activate a greater number of motor units (e.g., type II fibers) [79]. Moreover, other types of training are also suggested to increase LEMS strength, stabilize knee, leg, gluteal, and ankle muscles, and even prevent falls, such as multitasking exercises [80,81], square-stepping [82], exergames [83], salsa dance [84] and dual-task training [85,86].

Finally, our findings are consistent with the idea that, in advanced age, higher PA levels can contribute to a better perception of HRQoL [35,36,41,43]. In this context, having adequate control of static and dynamic BB and moderate to high levels of LEMS helps older adults feel more secure and confident in performing their ADLs. This is due to decreased possibility of facing imbalances and fear of falling, thus followed by increased confidence in balance [80]. In turn, all this increases the person’s degree of independence [51]. Concern about falling is a factor that affects the HRQoL of the older population [87], limiting mobility and restricting the individual’s social life. A possible explanation for the increase in the perception of the twelve health-related dimensions of HRQoL (physical and mental components) mediated by increased performance in LEMS and BB performance. This in turn, also implies an increase in PA levels, is that the three variables (LEMS, BB, PA) can act in the opposite direction to the gradual deficits caused by the vulnerable aging process, benefiting well-being [88,89]. Among the strengths of our study, we can mention the association approach (mediating role) of LEMS and BB in the relationship between PA and HRQoL in a large sample of community-dwelling older adults. Therefore, we offer quantified information on the mediating role of these two variables.

Our study includes some limitations. First, due to the cross-sectional design, causal associations were limited. However, our findings could broaden the understanding of the potential mediating role of LEMS and BB in the relationship between PA and HRQoL, which could stimulate future investigations. These need to adopt longitudinal approaches to detail the mechanisms of the study variables during aging. We also encourage future investigations focusing on the association between PA and HRQoL, mediated by LEMS and BB, including, for example, gender and/or age as potential moderators to determine differences. A second limitation is that the general findings may have been influenced by the nature of the sample (individuals from the island of Madeira). Therefore, the results should be generalized with caution and wait for replications with other samples.

## 5. Conclusions

These findings suggest that the relationship between PA and HRQoL among older adults is partially mediated by both LEMS and BB performance. We also show the role that LEMS plays in the control of upright posture (e.g., static and dynamic balance), as well as the proportion of the total effect of PA on HRQoL mediated by LEMS and BB. In this context, the serial mediation pathway model suggested that PA (e.g., regular physical training) may play a crucial role in counteracting the age-related decline in physical functions (LEMS and BB), consequently benefiting the perception of HRQoL. In turn, our findings also suggest that the performance of both variables (LEMS and BB) can provide useful information to clinical professionals about identifying and monitoring PA levels and HRQoL in the older population.

## Figures and Tables

**Figure 1 jcm-11-04857-f001:**
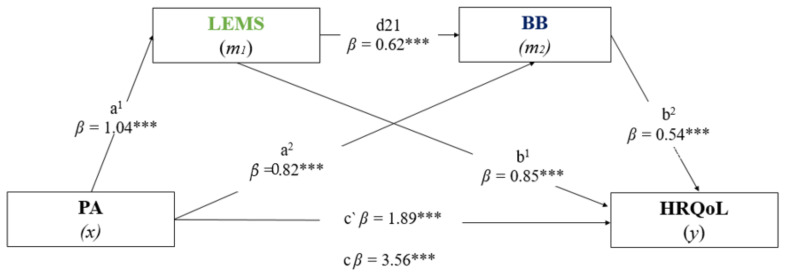
Analysis of parallel mediation of the effects of PA (physical activity) on HRQoL (health-related quality of life) through LEMS (lower extremity muscle strength) and BB (body balance). The analysis was based on 5000 bootstrap samples. The indirect effect was statistically significant at the 95% confidence interval (CI) when the CI did not include 0. Betas (*β*) were reported as the product of simultaneous regression with bootstrap replacement: (1) Path a^1^ and a^2^ = association between PA with LEMS and BB, respectively, (2) Path d21 = association between both mediations variable (*m*_1_ and *m*_2_), (3) Path b^1^ and b^2^ = association between LEMS and BB with HRQoL, (4) Path c, = direct effect (*x*–*y*): associations *m*_1_ green and *m*_2_ blue = indirect effect (*x*–*y*) by LEMS and BB, respectively. *** *p* < 0.001, c = total effect; c’ = direct effect; a = path Model 1; b = path Model 2.

**Table 1 jcm-11-04857-t001:** Main characteristics of the sample.

Variable	Full Sample (n = 802)
Age (years)	69.8 ± 5.6
Sex n (%)	
Female	431 (50.0)
Falls (n)	0.83 ± 1.60
Medication (n)	3.57 ± 2.56
Height (cm)	159.05 ± 8.69
Weight (kg)	74.77 ± 13.06
BMI (kg/m^2^)	29.51 ± 4.34
Hypertension	408 (50.9)
Visual impairment	489 (61.0)
Hearing problems	198 (24.7)
Musculoarticular problems	46 (5.7)
PA (n)	7.30 ± 1.23
HRQoL (n)	68.57 ± 17.96
LEMS	13.63 ± 4.14
BB	30.53 ± 7.40

PA: physical activity; LEMS: lower extremity muscle strength; BB: body balance.

## Data Availability

The data presented in this study are available upon request from the corresponding author.

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
