# Peer review of "Muscle Strength and Balance as Mediators in the Association between Physical Activity and Health-Related Quality of Life in Community-Dwelling Older Adults"

_jcm, 2022, doi:10.3390/jcm11164857_

Round 1

Reviewer 1 Report

This study aimed to examine in-depth whether lower extremity muscle strength and BB mediate the 119 relationships between physical activity and health-related quality of life in a large sample of community-dwelling older adults. The measured PA with Baecke questionnaire, HRQOL with a local native checklist, BB with FAB scale, and LEMS with SFT test. They concluded that the relationship between PA and HRQoL among older adults is partially mediated by both LEMS and BB performance. They also mentioned that the performance of both variables (LEMS and BB) can provide useful information to clinical professionals about the identification and monitoring of PA levels and HRQoL in the older population.

As I mentioned before, this is an interesting paper investigating the relationship between LENS and BB in the relationship between PA and HRQol. The title was original and novel and the number of objects is enough. The paper is well written and clear to read for scientists and also general readers. The method and material are clear and describe details well and the conclusion is consistent with the aim of the study and represents the results. 

Author Response

Dear Reviewer, we are very grateful for the overall positive review.

Best regards!

Reviewer 2 Report

This article revealed that the performance of both variables (LEMS and BB) provides useful information on PA levels and HRQoL in the older population by a community base population method.

The authors analyze with multiple mediators mediation analysis which needs to judge whether this mediating effect is statistically significant (different from zero). For this, there are two main approaches: Sobel test and bootstrapping. Would the authors provide more evidence from the literature about this statistical method, especially considering important factors such as low limb function or age? 

The disadvantage of the article is that the study design does not provide the category difference after interventional treatment. And the conclusion was well known with new acknowledgment. There is no characteristic classification and analysis, for example, gender, age group, different BMI status, or sarcopenia.

Author Response

Dear Reviewer, we are grateful for all the comments, and are available for future clarifications and/or corrections.

* Changes were made in the text using Microsoft Word's built-in track changes function.

1) The authors analyze with multiple mediators mediation analysis which needs to judge whether this mediating effect is statistically significant (different from zero). For this, there are two main approaches: Sobel test and bootstrapping. Would the authors provide more evidence from the literature about this statistical method, especially considering important factors such as low limb function or age?

Reply

Dear Reviewer, in the Statistics section we have added requested information on (1) Mediation Analysis, and also (2) to enrich the presentation of the mediation analysis, we also included previous studies focusing on the area of aging, which used this analysis to assess lower limb strength, quality of life, and physical function.

2) The disadvantage of the article is that the study design does not provide the category difference after interventional treatment. And the conclusion was well known with new acknowledgment. There is no characteristic classification and analysis, for example, gender, age group, different BMI status, or sarcopenia.

Reply

Dear Revisor, we agree with the reviewer about the study design. For this reason, we added in the limitation section that causal associations were limited due to the cross-sectional design. Also, in the conclusion section, due to this significant limitation, we always moderate our speech by saying that these findings suggest that the relationship between PA and HRQoL among older adults was partially mediated by both LEMS and BB performance. Also, we were careful in saying that the serial mediation pathway model suggested that PA may play an important role in counteracting the age-related decline in physical functions. Finally, we also indicated that the performance of both variables analyzed (LEMS and BB) could provide valuable information to clinical professionals about identifying and monitoring PA levels and HRQoL in the older population. It is recognized that longitudinal data are needed to clarify and sustain those established relationships.

We also agree with the reviewer that integrating characteristic classification and analysis, such as gender, age group, BMI status, or sarcopenia, will identify important variables that may affect the relationship between physical activity and health-related quality of life. We acknowledged that future research should address this interesting idea. However, in this study, we aimed to examine whether lower extremity muscle strength and body balance mediate the relationship between physical activity and health-related quality of life in a large sample of community-dwelling older adults.
